# Big Data Analytics in Sustainable Supply Chain Management: A Focus on Manufacturing Supply Chains

Joash Mageto

Department of Transport and Supply Chain Management, University of Johannesburg, P.O. Box 524, Johannesburg 2006, South Africa; joashm@uj.ac.za

**Abstract:** Sustainable supply chain management has been an important research issue for the last two decades due to climate change. From a global perspective, the United Nations have introduced sustainable development goals, which point towards sustainability. Manufacturing supply chains are among those that produce harmful effluents into the environment in addition to social issues that impact societies and economies where they operate. New developments in information and communication technologies, especially big data analytics (BDA), can help create new insights that can detect parts and members of a supply chain whose activities are unsustainable and take corrective action. While many studies have addressed sustainable supply chain management (SSCM), studies on the effect of BDA on SSCM in the context of manufacturing supply chains are limited. This conceptual paper applies Toulmin's argumentation model to review relevant literature and draw conclusions. The study identifies the elements of big data analytics as data processing, analytics, reporting, integration, security and economic. The aspects of sustainable SCM are transparency, sustainability culture, corporate goals and risk management. It is established that BDA enhances SSCM of manufacturing supply chains. Cyberattacks and information technology skills gap are some of the challenges impeding BDA implementation. The paper makes a conceptual and methodological contribution to supply chain management literature by linking big data analytics and SSCM in manufacturing supply chains by using the rarely used Toulmin's argumentation model in management studies.

**Keywords:** sustainable supply chain management; big data analytics; Toulmin argumentation model

## 1. Introduction

Sustainable supply chain management (SSCM) has received much attention in the decade ending in 2020 due to an increased awareness of climate change and environmental and social issues across the globe. SSCM requires firms across a supply chain to report not only on profits but also on environmental and social performance [1–3]. To achieve the objectives of SSCM, firms should set long-term goals on sustainability, be transparent in their reporting, develop a culture of sustainability and manage supply chain risks appropriately [1]. With globalization, manufacturing supply chains have become complex, making it difficult to monitor the sustainability practices of every firm especially in developing countries where there are no existing regulatory frameworks. Proper monitoring of sustainability practices can help avoid the over-exploitation of the environment and vulnerable communities, especially in developing countries. The complexity of manufacturing supply chains implies that some firms might get away with a socially and an environmentally irresponsible behaviour undetected especially in issues such as child labour [4]. However, achieving sustainable goals and avoiding the reputational and supply disruption that risks firms are likely to incur for participating in such a supply chain calls for the application of advanced technologies to collect, analyse and share relevant data from all firms in a supply chain to monitor their activities and take corrective action. The emergence of fourth industrial revolution (4IR) technologies, such as the Internet of things, artificial intelligence,

additive manufacturing, advance robotics and other advanced information communication technologies, has allowed enterprises to collect massive amounts of data across a supply chain on the three areas of sustainability [3]. The huge amounts of data collected are known as big data and can help minimize the complexity of monitoring sustainability practices in manufacturing supply chains. Big data analytics (BDA) technologies have the capability to continually analyse data and share results in real time to help generate new insights and make decisions to solve complex problems across the supply chain. Despite the fact that BDA allows the use of large-scale data to make evidence-based decisions [4–6], its use and likely effect on SSCM in manufacturing supply chains remains scanty.

Firms that implement SSCM are likely to reduce operational costs, improve their image and monitor their environmental actions by pursuing sustainability goals [7]. In addition, data driven decisions are likely to increase the effectiveness of not only the individual enterprises but also their supply chains in the implementation and monitoring of SSCM goals [3,8]. The application of BDA is likely to result in improved operational performance through accurate and faster decision making within the supply chain, especially when there is access to real-time data [7,9,10]. BDA can drive efficiency and effectiveness in supply chain management in terms of improved demand management, faster new product development, better supply chain risk management, better supplier management and development of efficient and robust supply chain designs, thus supporting the sustainability agenda [8].

Many researchers have examined the concept of big data analytics from various perspectives. For instance, the development of BDA capabilities in firms [8], BDA applications in SCM (Nguyen et al., 2018), BDA applications to logistics [11], BDA applications in raw materials flow in manufacturing operations [12], BDA and decision making in SCM [13], the effect of BDA capabilities sustainable supply chain performance outcomes such as sustainable product development (Jose et al., 2020) and dimensions of big data databases [14]. While many of the aforementioned studies have investigated BDA and SSCM separately, the current research links the two concepts in a proposed model, highlights the likely effect of BDA on SSCM and identifies the elements of BDA, thus bridging the literature gap. The study also makes a methodological contribution by applying the rarely used Toulmin's argumentation model in supply chain management. The findings of this study will help managers to select the right BDA tools to help monitor supply chain activities for SSCM. The following research questions are answered in this study:

1. What are the elements of big data analytics?
2. How does big data analytics enhance SSCM in manufacturing supply chains?
3. What are the challenges that inhibit the implementation of big data analytics in manufacturing supply chains?

The rest of the paper is organized into a literature review on sustainable supply chain management, manufacturing supply chains and big data analytics. Thereafter, the methodology, discussion and conclusion are presented.

## 2. Literature Review

### 2.1. Sustainable Supply Chain Management

Sustainability refers to the integration of environmental, economic and social goals to meet current needs without compromising the needs of future generations [15,16]. On the same note, sustainable supply chain management (SSCM) is the strategic and transparent management of supply chain activities by integrating the sustainability (environmental, social and economic) goals in all processes to meet stakeholders' requirements [1,2]. By integrating the sustainability goals in supply chain activities, a firm pursues not only the economic dimension but also social and environmental goals, also known as the triple bottom line (TBL) approach [1].

In the current business environment, firms across all sectors are expected to be act responsibly by conserving the environment and being transparent on their activities even as they endeavour to grow profits [7]. Thus, firms that pursue the three goals of sustainability

across their supply chains are considered nonexploitative and likely to perform better than nonresponsible firms [2]. Firms that implement SSCM practices seek to balance the economic, social and environmental investments internally as well as across their supply chains.

The concept of SSCM has been examined in prior research; for instance, [2] proposed a four-factor model whereby transparency, risk management, strategy and culture were identified as the four dimensions of SSCM. The strategy dimension calls on firms to incorporate sustainability in their corporate strategy. Instead of pursuing sustainability as an independent objective, better results are obtained when it is integrated into the strategy of the firm, and the benefits can be felt across the supply chain [2]. Therefore, the strategic objectives of the firm should be integrated with TBL goals to maximize return on investments through sustainability [17]. Monitoring the implementation of the dimension across the supply chain requires the close monitoring of operational level activities to ensure that they are directed towards not only economic but also environmental and social achievements [7]. The monitoring can be efficient and effective with big data analytics technologies [7]. Thus, big data analytics is likely to support the strategy dimension of SSCM.

The risk management dimension focuses on protecting the organization from hurtful disruptions that prevent the delivery of raw materials and subsequent distribution of goods and services to the final customer, which might result in a damaged reputation or even litigation [1,2]. Risks aforementioned can stem from a firm's products causing harm, producing environmental waste from the firm or endangering worker and public safety when handling the firms' activities [2]. Disruptions also account for likely risks and might be a result of heavy traffic along the supply routes, natural disasters, pandemics such as COVID-19, scarcity of raw materials and hurtful laws, regulations and policies [2]. Other sources of supply chain risks include poor demand prediction techniques, poor demand coordination within the supply chain, increased prices, noncompliance to quality standards and poor social and environmental performance [2]. The risks identified inter alia touch on all three aspects of sustainability; firms should identify activities that are likely to turn into risks and have contingency plans to mitigate supply-related disruptions and image risks [2]. Firms that engage in sustainable practices proactively are likely to lower supply chain risks that can disrupt commercial activities [18]. This implies that risk management is part of sustainability; thus, firms should build resilience and agility to ensure effective response and recovery to mitigate reputational and commercial risks [2]. Therefore, detecting of supply chain risks early and taking corrective action is only possible if firms collect and analyse data in a continuous manner—that is, if they apply big data analytics technologies to draw new insights and make effective business decisions.

The transparency dimension requires firms within the supply chain to share all relevant information through upstream and downstream linkages to ensure that all activities performed adhere to sustainability goals. Information flows from one location to all parts of the world in seconds driven by the internet via social media platforms such as Twitter and Facebook can reveal the unethical conduct of a firm instantly to a global audience [2]. This implies that if a firm is involved in unethical behaviour or sources from unethical suppliers, the information will be available online and may result in reputational or legal damages. Thus, firms are called to act sustainably not only internally but also across the supply chain and share all the relevant information with their supply chain partners [14]. Firms should also report to their stakeholders the sustainability initiatives they are engaged in (for example, green production activities), so as to create awareness [19]. Transparency, as a sustainability dimension, can be enhanced when firms embrace vertical as well as horizontal coordination across the supply chain. This might include adopting common sustainability standards and auditing procedures, which might result in low transaction costs compared to when this is done by individual firms [2]. Transparency is an important facet of sustainability as it enhances collaboration among the supply chain partners [2], allowing for trust and cooperation [20] as well as sharing of data [20] that can be used for

predictive modelling and decision making. I argue here that big data analytics can create the level of transparency required to foster cooperation. This is because BDA requires firms in a supply chain to share all relevant data regarding sustainability, perform data analytics and provide insights that are shared across the supply chain for effective decision making [14]. The sharing of these insights enhances transparency and encourages members to pursue sustainability goals with clear outputs [13]. Thus, BDA supports the transparency dimension of SSCM.

Sustainability goals cannot be achieved if the organizational culture is not supportive. Savitz [21] claims that TBL goals should be embedded in the organizational culture to drive sustainability internally and across the supply chain. This implies that organizations should have sustainability and ethical conduct as part of their core values. Similarly, Carter and Rodgers [2] advise that corporate culture should be connected to environmental and social activities that the organization is involved in. In addition, supply chain activities outside the core values (sustainability being one of them)—for example, sourcing from suppliers who use child or forced labour—should be identified as potential sources of risks and corrective action should be taken. Developing a sustainability supportive culture and monitoring the same across the entire supply chain remains a complex issue. However, the 4IR technologies, such as big data analytics, offer opportunities for firms to share data, which can be analysed continuously using advanced analytics to develop insights that can help firms make the right decisions regarding sustainability.

The four dimensions or facets of sustainability as referred to by [2] demonstrate that risk management can be improved when there is a balance between environmental and economic goals. On the other hand, transparency requires a balance between economic and social goals, as environmental and social goals are ingrained in the corporate strategy and organizational culture of the firm. Therefore, to achieve sustainability, it is important that a firm seeks a balance between the three goals with clear measurable metrics across the four facets. The concept of sustainability in has been addressed by many researchers and academics, signalling its importance in the contemporary business environment [16,22,23]. However, there is a limited number of studies that have examined whether big data analytics has any influence on the SSCM practices of manufacturing supply chains. This remains the focus of this study.

### 2.2. Manufacturing Supply Chains

Supply chain management (SCM) refers to the planning, organising and directing of all activities involved in the movement of raw materials from suppliers to manufacturers, distribution of finished goods to customers and management of relationships with all the supply chain partners, including suppliers, logistics service providers, information technology providers and customers [1]. In manufacturing supply chains (MSCs), especially in cases of large manufacturers, the manufacturer becomes the centre of the supply chain with upstream and downstream linkages [24]. Upstream linkages are characterized by raw materials acquisition, while at the manufacturing site, prominent activities include product development, engineering and production. Downstream activities include distribution, use and services as well as product disposal [24].

MSCs are faced with challenges of meeting ever-changing customer requirements while ensuring sustainability in terms of social, economic and environmental aspects. The 4IR offers technologies (such as big data analytics) and thus an opportunity for MSCs to meet the three goals of SSCM, which include environmental, economic and social. Big data analytics offers an opportunity for manufacturing supply chains to implement smart factory and smart logistics concepts, whereby smart data is collected by smart devices across the supply chain from raw material acquisition to the end of life of a product. Data analytics are applied to help draw new insights for better business decision making along the three sustainability dimensions. Figure 1 illustrates a smart and sustainable manufacturing supply chain in the 4IR era. Figure 1 reveals that data is collected synchronously from every member of the supply chain and stored in a cloud. The data is then subjected to

analytics, and results are shared synchronously to all members of the supply chain for quick and accurate decision making. This means that the smart factory will schedule production based on real-time demand data from the retail stores and orders from smart homes/customers. The raw materials requirements can also be reflected at the supplier point automatically, and the manufacturer can see the likely delivery date. Any disruption along the supply chain is known instantly, and mitigation measures are employed to avoid damages. The application of 4IR technologies, such as smart factory, procurement 4.0, transport track and trace and BDA, results in reduced emissions and, to this extent, promotes SSCM of MSCs.

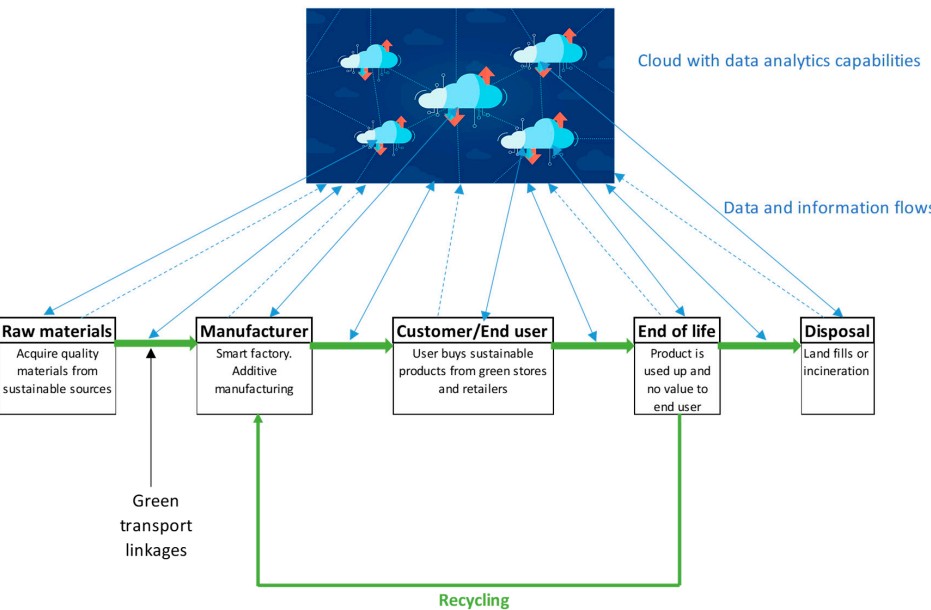

**Figure 1.** Smart data collection across a manufacturing supply chain.

Prior research [9,13,25] argues that one way of developing sustainable supply chains is by leveraging on big data collected from each of the role players. Smart devices can collect smart data continuously and apply advanced analytics on the big data, and the resulting insights can advance knowledge and guide business decision making, especially on how to achieve sustainability [24].

Data is collected across the manufacturing supply chain operations, including raw materials acquisition, manufacturing, use and services and end of life (Figure 1). Therefore, we argue that big data analytics makes use of intelligent or smart devices and technologies, which allow for continuous data collection and analysis and for drawing insights for better decision making [24].

Smart logistics requires that the smart devices have capabilities to respond to unforeseen events, such as rerouting in the case of traffic on the main route to reduce distance and fuel consumption. Within the manufacturing warehouses, automated guided vehicles can be used to pick and store goods and radio frequency identification tags and quick response codes can be used for ease of identification and tracking of goods to create visibility and inventory optimization [24]. The use of the smart devices creates efficiencies and transparency within the supply chain, thus promoting SSCM sustainability, especially in terms of the economic and environmental dimensions that advocate for the efficient use of resources to increase profits while caring for the environment.

MSCs are expected to be sustainable so as to reduce costs and improve a company's image to stakeholders [26]. Sustainable manufacturing includes pollution prevention and product stewardship. Pollution prevention includes activities related to the reduction of energy and raw materials usage and operational and solid waste reductions. Product stewardship includes activities such as creating awareness through sustainability education,

measurement and reporting, product redesign, use of new management systems and investments in research on sustainable products [26].

Some manufacturing firms have embraced the concept of sustainable supply chain management by implementing a number of initiatives at various levels of the supply chain, including supplier selection, logistics, product design and development, customer service and reverse logistics [27]. The initiatives are supported by green purchasing, green manufacturing, environmental collaboration with suppliers and sustainable reverse logistics [22]. Sustainable manufacturing firms should also focus on optimal resource utilization, reducing the use of hazardous materials and recycling to achieve sustainability goals [22,28]. The SSCM initiatives can result in the improvement of operational as well as supply chain performance [2,22] and allow for gaining a competitive advantage [27]. The challenge is that when implemented by a single firm (for example, a manufacturer within the supply chain), sustainability might create little or no impact; thus, it should be embraced across the supply chain for maximum benefits. This implies that suppliers of raw materials should be socially and environmentally responsible, as should retailers, logistics service providers, information technology service providers and all stakeholders. Therefore, I argue that when data is collected across the supply chain on these activities, the analytics of these data will provide new insights that can help promote the sustainability of manufacturing supply chains.

BDA technologies allow manufacturing supply chains to improve materials management by providing relevant information relating to type of materials, quantities, location within the supply chain and likely delivery dates. This helps supply chains to hold precise inventories and better control, make use of the just-in-time philosophy and reduce freight costs [4]. The application of BDA technologies can help eliminate wastage from supply chain management operations, monitor productivity and performance and allow for automation [4]. BDA can help monitor data related to servicing and provide insights on likely accidents from machine failure. BDA provides insights regarding asset utilization and inventory optimization to avoid wastage. Insights of quality of materials and goods across the supply chain are obtained from BDA to ensure competitiveness and entry into global markets. Demand forecasting using real-time data is possible, and it helps to reduce costs while increasing service levels to customers, thus increasing company profits. New product development milestones are achieved within shorter periods given that all supply chain partners are sharing data and new insights for better decision making are generated continuously from the analytics.

In terms of the social aspects, BDA can help with remote monitoring of labour issues, including compensation, likely use of forced or child labour and exploitation of communities where the supply chain facilities are located. BDA can monitor and report on energy consumption and emission of effluents across the supply chain. BDA monitors supply chain members and activities that consume resources over and above the recommended safety standards for purposes of compliance [7]. Suboptimal processes within the supply chain which might be consuming resources are identified through the analytics, and the right decisions are made for optimization to achieve environmental goals [4]. Ammar [4] argued further that BDA's real-time data analysis ability results in better quality monitoring, better reports, transparency, smart alerts, production efficiency and better manufacturing supply chain planning. Therefore, BDA supports all the aspects of sustainability of manufacturing supply chains by using smart devices to collect, store and analyse data continuously for decision making.

### 2.3. Big Data Analytics

Big data is characterised as high-volume, high-velocity, and high-variety data assets [8,29] commonly known as the "5Vs" [30]. This implies that the concept of big data comprises massive amounts of data that increase rapidly, as do the heterogeneity of the data and their sources [4]. Researchers have further argued that it comprises data mining, storage and visualization. The authors of [30] claimed that big data constitutes complex data sets,

generated continuously, and requires innovative analytics. The authors of [29] observed that big data requires more advanced software and database capabilities when compared to traditional datasets. Therefore, big data is associated with the creation of huge amounts of data of different types from various sources and sharing the data at high speeds to various points where it is required for decision making [8]. The big data described hitherto should be managed and analysed to provide new insights for decision making. The process of examining big data to identify patterns and relationships using advanced technologies to develop predictive models and complex statistical algorithms, including machine learning for informed business decision making, is referred to as "big data analytics".

Successful management of big data and associated analytics requires the right technologies, human resource skills and high-quality data [4]—this results in the "2Vs" of data analytics, that is, veracity and value [31]. Veracity refers to the data quality and trust levels attached to the data sources [32], while value relates to the rigorousness associated with data analytics to uncover valuable insights to support decision making [31]. Therefore, big data analytics is fully described by the "5Vs" of volume, velocity, variety, veracity and value. Big data analytics can help organizations identify ways to reduce inventory costs, reduce communication gaps, monitor equipment based on the data it transmits and provide transparency across supply chains [4]. This is because BDA makes use of state-of-the-art analysis technologies to glean valuable knowledge from big data to facilitate data-driven decision making [4].

In a big data environment, huge amounts of data are generated from many sources, including social media, online shopping sites, search engines such as google, smart sensors and devices and shared in real time with supply chain partners. Analytics are applied to the data generated to draw insights into business, transport and supply chains, among other sectors. As such, big data analytics makes use of advanced technologies to analyse the big data to create valuable information for business decision making [4]. The analytics applied to the big data include data mining, machine learning, social network analysis and data visualizations. The analytics help identify opportunities to reduce costs, create a better understanding of supply chain activities, allow for integration of data from all supply chain partners and have the potential to develop higher competitive advantages.

Big data analytics requires more advanced software, higher efficiency and advanced data management technologies to handle both structured and unstructured data generated in a big data environment [30]. With the advanced technologies, BDA offers various capabilities, such as analysing consumer activities on online marketplaces in real time and helping in decision making; can handle both structured and unstructured data and can enable firms to reduce costs to gain global competitiveness [30]. Generally, the activities involved in BDA can be identified as including the development of objectives, goals and policies which must aligned to SSCM practices. The other activities include data acquisition, management, storage and processing and the application of analytics. After analysis, there should be visualization and reporting, sharing and publication to create awareness to stakeholders.

Big data analytics can use descriptive, prescriptive or predictive levels. Prescriptive analytics can recommend, for example, the adoption of an optimization, such as route optimization in the case of transportation. Predictive analytics might apply classification algorithms in manufacturing, logistics, transportation, procurement and sustainability. The algorithms help identify whether a particular process or activity is sustainable or not. Descriptive analytics make use of association techniques to test relationships among items within the supply chain to support decision making [31].

### 2.4. Dimensions of Big Data Analytics

Research on the deployment big data analytics in supply chains is in its nascent years and likely to grow as organizations adopt it to develop demand-sensing networks based on the data generated both internally and externally. Previous studies on BDA have focused on a variety of issues; for instance, [14] studied its dimensions, although their focus

was on big, open and linked government databases which are generally different from private enterprises and supply chains. They claimed that public open and linked big data analytics should have aspects such as the economical, technical viability, engagement with general public, social awareness, proper laws to protect entities and transparency from the government regarding how the data is used. Research conducted by [33] implicitly identified BDA to have analytics and reporting dimensions; hence, it is still inconclusive what the actual aspects of BDA are. Research focused on identifying the dimensions of BDA is limited. This creates a need to identify BDA dimensions that relate to private entities and supply chains. In this study, we have gleaned literature to identify six elements of BDA.

A BDA ecosystem should have intelligent data processing capabilities. Data processing can be achieved if the BDA environment has intelligent devices and software with capabilities to collect data from various sources and forms, process it into meaningful information automatically and export it for reporting [34,35]. The data processing is expected to be in real time, continuously and from different perspectives, providing insights that help drive the sustainability objectives across supply chains [33,34]. Data processing should be coupled with supply chain wide transparency, especially during data collection, to encourage supply chain partners to share relevant data on the activities they are engaged in and the likely effects to the environment and community [36]. Transparency here refers to the accountability and openness of all the supply chain partners to advance the goals of SSCM [36]. Transparency is likely to thrive on the ethics and legitimacy of the BDA concept to the supply chain partners. Transparency across the supply chain can be achieved if the members agree to collaborate [37]. The areas of collaboration and data sharing that promote SSCM objectives include sourcing, production and distribution activities [35]. Transparency requires that each of the members be ready to share all the legitimate data that can be processed to help monitor SSCM practices internally as well as externally. Accurate data processing can be achieved when the supply chain partners improve their technical abilities through the establishment of an ICT framework on business processes, software technologies and collecting data that can support the BDA and SSCM goals [14,38]. The framework should allow for resource sharing and cost reduction, in addition to being compatible with all supply chain member systems and processes [39]. The desired ICT infrastructure for effective data processing should be secure, flexible and cost-effective [40]. In addition, the data collected from various sources and formats should be relevant to SSCM metrics, which are economic, environmental and social [35]. Therefore, data processing is an important dimension of BDA.

An effective BDA should have reporting capabilities. The reports should be relayed at near real time to the users or supply chain partners in formats that can help them make decisions or draw insights. It is expected that the reports will be relevant to an organization within the supply chain when received in a timely manner [33,41]. The BDA reporting should promote engagement among the various supply chain stakeholders to achieve cooperation where necessary [41,42]. SSCM requires stakeholders to promote activities that enhance sustainability. The stakeholders should develop interest in applying insights drawn from BDA reporting to promote sustainability [14]. Reporting helps to identify conflicting stakeholder interests and align them with sustainability goals. Furthermore, [33] highlights that engagement in SSCM issues identified from the insightful BDA reports enhances partnerships on innovation and new product development to promote sustainability practices across the supply chain. The reporting capabilities of BDA should promote the achievement of the social objectives of communities where the supply chains operate [14,42]. For instance, the community should expect reports on capacity building programmes, literacy rates, education and skills development to reduce inequalities and create sustainable communities into the future. This implies that supply chain partners must be accountable by allowing industry regulators to use the BDA reports to institute control measures directed towards improving SSCM [14]. In addition, [43] argues that reporting promotes the transparency and ethical conduct of supply chain members and thus

improves SSCM. Therefore, the reporting features of a BDA system should be enhanced and improved to meet the needs of not only the enterprises but also all stakeholders to promote the sustainability of supply chains. Generally, the reporting capability should be able to provide reports on enterprise and supply chain performance on economic, environmental and social aspects to make it easier to identify the weak links regarding SSCM. Therefore, reporting is an important dimension of BDA.

Effective BDA should have elaborate security features to secure proprietary data from unintended access and use [41]. The BDA system should also be secured from external attacks by hackers [41]. Data security also includes issues dealing with ownership of the data within the various nodes of the supply chain [44]. Security can be enhanced by having authentication details to access and extract information and by keeping a trail of user activities [38]. Data transmitted in the BDA system can also be encrypted to protect it from theft [38]. BDA should also have fraud analytics capabilities to identify security breaches and take corrective action. The security analytics should also be able to detect unethical practices by supply chain members that are likely to impede achievement of SSCM goals. To achieve effective security and safety of the BDA ecosystem, a legal framework to govern how the data will be used by the stakeholders is important regarding privacy [14]. BDA requires that supply chain members incorporate elaborate data privacy protocols to govern what and how data is collected, stored, used and reused [14,45,46]. Data privacy is an important aspect in BDA, and this requires all stakeholders to act ethically to build trust and encourage other members to report on sustainability. However, the laws governing data privacy and security should not limit transparency to stakeholders. Thus, the security dimension should encourage and assure confidence among the supply chain members to share data relevant for BDA so as to draw insights for SSCM-related decision making.

An effective BDA ecosystem should have analytics as one of the dimensions. Data analytics refers to the application of advanced technologies to perform complex analysis and modelling of structured and unstructured datasets to draw new insights for decision making [47]. The analytics include risk analytics, which include the prediction of future events that are likely to be sources of disruption and help with mitigation insights. Analytics also include automation of the decision-making process based on the analytics. Machine learning is one of the methods used to conduct analytics of both structured and unstructured data by finding patterns and helping to draw new insights for the benefits of supply chain partners [48]. Generally, analytics can be descriptive, predictive and prescriptive to help draw new insights from big data for decision making [38]. Therefore, analytics as a dimension can help improve the quality of decisions made by management from big data and more, especially regarding SSCM.

Big data and indeed BDA can only gain support from enterprises if they promise and deliver value to all stakeholders in a supply chain. Thus, the economic dimension of BDA is important for its adoption, especially in manufacturing supply chains. It is expected that insights obtained from BDA will help advance SSCM practices as well as the economic well-being of stakeholders [6] within the manufacturing supply chains. The anticipated economic gains include insights that result in better business decisions geared towards cost reduction and improved customer service [49]. The data that results in economic gains should be shared and the value obtained reported adequately to create awareness on SSCM [50]. Reporting of BDA-related economic gains can result in value to the enterprise, supply chain and community in terms of which activities consume costs and are not sustainable. Therefore, the economic dimension can guide enterprises to ensure that they launch a BDA ecosystem that is economically viable and can help to achieve the objectives of SSCM.

Finally, BDA should have capabilities to integrate data from various systems and enterprises within the supply chain [38]. BDA should promote interoperability, flexibility and communication between functions and supply chain partners [38,41]. Software technologies should be shareable and allow for integration across the supply chain for quick response and better decision making. The data infrastructure to support BDA should

be such that it allows for interoperability and integration with a variety of systems and adherence to certain quality standards [51]. A BDA ecosystem can add value, especially for purposes of SSCM, if it can offer data fusion and interoperability, which helps to combine data from varied sources [41]. The data infrastructure should also support data linkages and interoperability [14]. The interoperability and compatibility of complex data promotes the adoption of BDA by partners within a supply chain [5]. In addition, cooperation in a BDA ecosystem is achieved through interoperability and real-time data collection and sharing capability [52]. Therefore, the integration and interoperability dimensions are important for an effective BDA to achieve SSCM.

### 2.5. Big Data Analytics and SSCM

A supply chain is a complex network of interconnected suppliers, manufacturers, customers and service providers, such as logistics and information technology service providers. The complexity of a supply chain makes it difficult to monitor the activities of each of the supply chain actors [5]. It becomes even more difficult to monitor supply chain activities when pursuing a sustainability agenda, given that some supply chain partners may not be transparent. However, big data analytics offers an opportunity to analyse data from each supply chain member as well as other external sources, thus making it easy to monitor activities that may result in unethical behaviour and take corrective action [6,10]. This is because BDA has capabilities to integrate the various links of the supply chain and collect and analyse data to draw new insights that can help in supply chain planning and visibility decisions [29], as discussed in the previous section. BDA has been applied to manage supply chain issues such resilience, sustainability, risk management and agility [29].

Prior studies have examined the BDA concept and SSCM, albeit in a fragmented way, whereby the capabilities of BDA and facets of SSCM are studied separately. Thus, there is a need to identify the various links established between BDA and SSCM to help develop theory that will guide future research. BDA promotes social sustainability, as argued by [33], which focused on how BDA can be applied to identify social risks and promote social sustainability in the supply chain. The issues examined included workforce safety, security and health as well as unethical behaviour within a supply chain.

Prior studies have also examined BDA and the economic dimension of sustainability by utilizing models to optimize supply chain operations [29,33] and to predict the financial impact of every supply chain activity [47]. Big data analytics is applied to optimize SCM functions: for instance, production planning, transportation, warehousing and distribution and sourcing to identify opportunities for cost savings [30,31,53]. BDA has been applied in warehouse space optimization and supplier selection and management to minimize related costs and improve efficiencies [33]. BDA has also been applied to coordinate the supply chain as a whole, especially in logistics planning, tracking goods on transit to optimize inventory levels, quality control in manufacturing, demand sensing and shaping to improve customer service [30,54,55]. To achieve the desired economic efficiencies, BDA uses advanced technologies, such as smart sensors, QR and barcodes, radio frequency identification tags and the Internet of things to integrate and coordinate supply chain linkages for ease of continuous data collection, smooth flow of goods and accompanying data and information [11,38]. Thus, BDA helps to achieve the desired cooperation and collaboration and quick responses to the market, making the specific supply chain more competitive than non-BDA supply chains [7,55].

Environmental sustainability can also be achieved by leveraging BDA to reduce delivery time through the direct connection of millions of customers, which is made possible by real-time information sharing, thus reducing amounts of energy consumed [30,32]. BDA also helps monitor activities of supply chain partners to identify unsustainable or unethical activities or any other environmental misconduct [7,33]. Furthermore, [7] argues that BDA can help to obtain an accurate impact of specific economic activities on the environment by monitoring relevant indicators.

The application of BDA is likely to benefit supply chains to achieve sustainability in the social dimension by reducing the supply chain risks associated with the procurement of goods and services, especially from global markets [31,55]. In addition, adopting BDA is likely to integrate supply chain activities, thus promoting transparency and sharing of information [7], making it easier to detect unethical practices that may impact the communities negatively. BDA also encourages a collaborative and cooperative culture which results in the ethical conduct of supply chain partners [5]. BDA provides a platform to compare past environmental conduct with the present to help forecast future social problems [7]. Therefore, BDA has recorded benefits for all the three dimensions of sustainability; its adoption is likely to promote sustainability.

### 2.6. Challenges of Big Data Analytics Implementation

Data is increasing rapidly but cyberattacks and data privacy issues are increasingly becoming important, thus creating a barrier for firms that want to implement BDA [56]. However, continued development and innovation in ICT offers opportunities to develop more robust BDA tools to analyse the data generated by manufacturing supply chains while maintaining the required security measures.

Although big data analytics has many benefits of promoting sustainability, especially in manufacturing supply chains, there are many barriers to the implementation of these technologies. Some of the barriers include technology, lack of human resources, and complexity in data integration due to security, privacy and policy issues [30,56].

## 3. Methodology

A research methodology describes the theories, methods and procedures used to help solve the problem at hand. The current study followed Toulmin's model of argumentation [57]. The research comprises a series of arguments, whose basic parts include a claim, grounds and a warrant [57–59]. The three parts are indispensable to an argument [60]. An argument begins with a claim or an assertion to be proven. For the claim to hold, there must be grounds, evidence or facts in support [59]. A warrant acts as a moderator between the claim and the grounds by linking the two. In addition to the three, there is a backing, a rebuttal and a qualifier, and these three are secondary to the earlier three [59,60]. In this paper, a review of relevant literature is conducted and discussions and interpretations are made based on the Toulmin's argumentation model. Therefore, this paper follows an interpretivist philosophical orientation and a qualitative research approach to answer the research questions presented under the introduction.

### 3.1. Selection of Literature and Analysis

Literature related to sustainable supply chain management, big data analytics and manufacturing supply chains were searched in Emerald, Science Direct, Wiley Online, Taylor Francis and Sabinet databases. The key words used for the searches were "sustainable supply chain management", "big data analytics", "big data" and "manufacturing supply chains". The identified articles were screened, and only peer-reviewed articles were selected. Further screening ensured that the articles were relevant to the research problem at hand. Duplicates were eliminated, and the resulting 46 articles were considered for this study.

The content of the articles was grouped according to the predetermined themes of SSCM, BDA and manufacturing supply chains. The literature review was performed as per the thematic areas identified. In addition to the mentioned themes, the challenges of BDA implementation and dimensions of BDA were identified. To establish the relationship between BDA and SSCM, the reviewed literature was interpreted in line with Toulmin's model of argumentation six-step process, clearly stating and supporting the claim and qualifier, grounds, warrant, rebuttal and backing. Some of the literature reviewed is summarized in Table 1; thereafter, the discussion is presented in Section 4. Figure 2 illustrates the methodology followed in this study.

**Table 1.** Key concepts addressed in prior BDA research.

| Author(s) | Key Concepts Addressed | Industry | Methodology | BDA/SSCM Dimensions Gleaned | Key Findings | Gap |
|---|---|---|---|---|---|---|
| [25] | Application of BDA to improve operational excellence and sustainable supply chain performance | Mining | Quantitative | Security | BDA management capability influences SSCM | Did not examine the dimensions of BDA or its effect on SSCM |
| [14] | Elements of big and open linked data analytics | Public sector databases | Qualitative; literature review | Economics, data privacy, technical, transparency, social and engagement | Essential elements of big and open data analytics in public databases were identified | Did not identify essential elements of BDA in private enterprises and supply chains, SSCM was not addressed |
| [24] | Applying industry 4.0 to achieve SSCM | Manufacturing | Qualitative; literature review | Data processing, integration, economic, social and environmental | Horizontal and vertical integration coupled with smart factory and end-to-end engineering can result in SSCM | The dimensions of BDA and effect on SSCM was not addressed |
| [38] | BDA and Internet of things | Not mentioned | Qualitative; literature review | Data processing, analytics, security, integration and reporting | Identified literature based on important parameters of IoT and BDA | Dimensions of BDA and SSCM were not identified |
| [56] | Addressing the barriers to BDA | 3PL, local government, car manufacturer and entertainment | Qualitative; literature review | Security breaches, culture, economic | Infrastructure readiness, privacy issues and a vision on big data can overcome the barriers | Link between BDA and SSCM was not addressed |
| [29] | Application of big data in SCM | Manufacturing | Case study/Delphi method | Integration | Big data analytics can reduce delivery time, inventory cost, operational cost and improve customer service through increased visibility | Dimensions of BDA not identified |
| [61] | BDA application in logistics and supply chain | Logistics | Qualitative; literature review | Analytics, integration, data processing | BDA has a wide application across logistics network and research is growing this area | Link between BDA and SSCM not explored |
| [62] | BDA application in supply chain relationship in banking | Banking | Case study | Security, integration | BDA can result in customer segmentation, product affinity | BDA dimensions were not examined |

| STEP 1 | Literature review | Literature search; identified 245 articles |
| | | Title screening (142) |
| | | Eligibility; abstract screening (53) |
| | | Included; peer-reviwed only (46) |
| STEP 2 | Application of Toulmins Argumentation | Set the Claim |
| | | Establish the Grounds |
| | | Establish the Warrant and Backing |
| | | Establish the Rebuttal and Qualifier |
| STEP 3 | Discussion and Conclusion | Address the research questions and present proposed model |

**Figure 2.** Research process.

*3.2. Steps in Toulmin Argumentation Model*

1. *Claim:* The first step in an argument is making a claim. A claim can be seen as a conclusion (Trent, 1968). In addition, a claim is a standard, an assertion or a thesis of the research [59]. Under this step the main question seeks to know the position that the author or proponent of an argument wants the readers to take. For instance, in this study, the claim is that the implementation of BDA can result in better SSCM, especially in manufacturing supply chains, which are usually complex and global and blamed for emissions.

2. *Grounds:* A claim can only hold if there is evidence, facts or grounds that support it. Grounds can also be in the form of logical reasoning or statistics. Grounds are opinions or citations from authority. The pertinent question at this point relates to what the author puts forward to persuade the reader to agree to the claim [60]. In the current study, evidence that supports the claim will be sought from published peer-reviewed academic research.

3. *Warrant:* A warrant certifies the claim as true based on the grounds or evidence. Warrants refer to the common beliefs within a particular discipline. They generally provide the reasons connecting the grounds to the claim. The pertinent question to ask here relates to what the connection between ground and the claim would be [59]. More specifically, the grounds support the stated claim since BDA has smart technologies that help collect and analyse relevant data that can result in optimized supply chain operations, reduction of emissions and prevention of unethical behaviour.

4. *Backing:* In cases where a warrant is not acceptable to readers as is, backers are introduced. Backers are items meant to certify the supposition of the warrant [60]. As such, backing supports the warrant. The pertinent question under this step relates to the reliability of the movement from grounds to claim. For instance, the warrant presented in (4) above is supported by data integration and the quality of data collected by the smart sensors that are part of BDA.

5. *Rebuttal:* In some cases, arguments face conflicting perspectives. These might be objections raised by readers or counterarguments. The objections should be addressed so as not to weaken the claim. A rebuttal may also refer to an alternative interpretation of evidence. A rebuttal recognizes that the claim might not hold under some situations and thus acts as "safety valve" [60] p. 45. The question in this step relates to the possibilities that might negate the arguments [59]. This is related to the likely reasons why BDA may not promote SSCM. For instance, there might be a relationship between BDA and SSCM but not necessarily a positive one.

6. *Qualifiers:* Arguments are about chances, not certainty. Qualifiers make a claim flexible. It uses words like "possibly" or "presumably" to qualify the claim.

The six steps of Toulmin's argumentation model can be expressed structurally as illustrated in Figure 3.

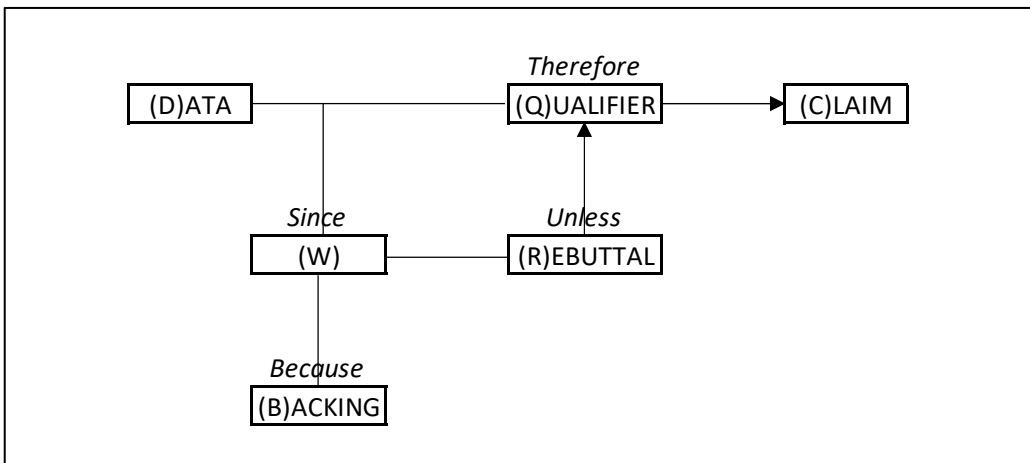

**Figure 3.** Illustration of Toulmin's argumentation model. Source: [60], p. 45. Data, Grounds; W, Warrant.

## 4. Discussion

Big data and data analytics are identified as "big data analytics" and characterized by the "5Vs" of volume, velocity, variety, veracity and value [30]. Data are identified as a means of integrating a supply chain. Data processing is identified as an important dimension of BDA; it includes real-time continuous collection of heterogeneous data using intelligent devices, modelling, data mining and analysis from various perspectives to provide new insights [34,38]. Insights from data analytics can aid more in sustainability-related decision making [35]. Prior literature [8] confirms that big data can only be collected across the supply chain and processed to develop insights for decision making if there is transparency in data processing. Transparency requires that the supply chain partners be open and accountable in terms of the quality of data that they share as well as the ethics of how the data is used [14]. Openness and accountability will allow for the supply chain members to make decisions collaboratively based on the insights from the data analytics. Effective data processing is also dependent on the right technical capabilities, which ensures that the right technologies and data infrastructure are in place to collect quality data, store and analyse it to obtain insights for decision making [35]. There should also be a data strategy that aims at meeting specific objectives for the individual firms and the holistic supply chain framework. The technical dimension provides guidance on the minimum technology and data infrastructure requirements a firm should possess to benefit from data processing capabilities.

The security dimension of BDA requires that the data collected and analysed should be protected from unintended use by supply chain partners and from external attacks [41]. The legal ownership of the data should be clearly set out so that conflicts are minimized [44]. As per the security dimension, the systems used should allow for authentication for access to the data and encryption when the data is transmitted within the network. BDA security capabilities should also include fraud analytics for early detection of unethical conduct. The security dimension should also help maintain clear security and privacy laws that govern how the data is collected, handled, analysed and reported for the benefit of all supply chain members [45,46]. A BDA ecosystem should have an intelligent reporting capability.

The reporting dimension provides real-time visualizations of the information from the data for decision making [35]. Reporting provides a basis from which regulators can be able to institute control measures [14] to improve SSCM in manufacturing supply

chains. Reporting should also promote engagement, which requires that the reports generated should be relevant in meeting the interests of stakeholders [33]. This implies that reporting should include the social aspects that supply chain partners are engaged in, such as capacity development, education and skills development [14,42]. The reports provide a window through which the various stakeholders will have access to information regarding the activities performed across the supply chain, its sustainability performance and some of the activities initiated by individual firms such as reducing carbon footprint and recycling [38]. The sustainability reports extracted from the analytics should be used to engage all the supply chain stakeholders with the aim of promoting sustainability, especially in areas of innovation, new product development and impact on the social wellbeing of the community [33]. The decisions made should meet the requirements of all stakeholders: the need for all the supply chain members to participate freely, collaboratively and cooperatively in BDA implementation at the various processes of the manufacturing supply chain. Big data analytics should be for the good of the communities where the firms and the supply chains operate. The reporting, especially on SSCM and associated insights, should be shared with the communities where the firms operate. In addition, there should be capacity building in the communities in terms of technical skills, and the BDA reports should be made available for reusability for the common good of society and all the stakeholders. The BDA should also report on the digital gap in society along the lines of literacy and education, so as to engage all the stakeholders to create equity. Despite the importance of quality BDA reporting, it can only be achieved when there is effectiveness in data processing and technical capabilities within the BDA ecosystem, which helps draw new insights that result in evidence-based decision making [38,42].

Analytics dimension is another dimension of BDA. Analytics make use of advanced software and algorithms to model both structured and unstructured datasets to help draw new insights [47,48]. One of the methodologies used to perform BDA analytics is machine learning algorithms, which can include predictive analytics and complex modelling [63]. The analytics can generally be classified as descriptive, predictive and prescriptive [38]. Predictive analytics are considered to be important as they can result in new insights that can improve supply chain visibility, resilience, cost savings and information transparency, which are important in SSCM [63]. In addition, [9] argued that predictive analytics can allow for supply chain integration and promotion of sustainability across the supply chain.

The Integration dimension of BDA allows for the seamless connectivity of various supply chain partners' systems to share data and subject it to analytics for generation of insights. Integration occurs when quality data is collected continuously from each of the supply chain partners and shared in real time [38]. Without integration, the insights or reports obtained from BDA might be suboptimal, as they will be based on a few cases and not from all partners. The integration should also be possible with external stakeholders, such as the government, so as to collect rich data. The authors of [38] argued that the integration dimension should support interoperability, flexibility and communication across the supply chain. Integration as a BDA dimension results in quick response, seamless compatibility of systems and processing of data from various sources and types to promote SSCM objectives [41,51]. Integration can only be possible when there is supportive information technology infrastructure across the supply chain.

Finally, the economic dimension requires that big data analytics be economically viable in terms creating value for the supply chain members as well as the various stakeholders. Specifically, big data analytics should result in operational cost minimization, thus supporting the study in [45]. Big data analytics should also create value to society by promoting sustainability goals for the common good, as also argued by [14]. To create value, supply chain members should be ready to commit financial resources to invest in technologies and data infrastructure, which are aligned to the economic, social and environmental objectives. This implies that big data analytics should not only be commercially enhancing but also environmentally and socially to promote sustainability across the supply chain. Therefore, the dimensions of BDA have been identified as data processing, security, reporting, analyt-

ics, economic and integration. These dimensions when maximized are likely to provide insights that promote SSCM goals across a given supply chain.

The relationship between big data analytics and SSCM in this study is examined using Toulmin's argumentation model. As per [57], an argument should have at least a claim, ground and warrant. In addition, the stated claim can be enhanced by adding a rebuttal and a backing [59]. To lay foundation for the *claim*, this study highlights that data is a medium of integrating the various links of supply chains through its collection and sharing across the supply chain. The data is expected to be quality for effective decision making across the supply chain (Brandenburg et al., 2016). Quality data can only be obtained when supply chain members are in a cooperative and collaborative transparent relationship with reasonable accountability. The application of BDA enhances the transparency facet of SSCM by creating high visibility across the supply chain, which promotes a win-win situation for all stakeholders [2]. Transparency minimizes unethical conduct and misreporting as well unnecessary bottlenecks, thus increasing efficiency in resource utilization, which eventually promotes sustainability. Thus, BDA enhances SSCM, as it encourages transparency through the sharing of data on processes and activities across the supply chain.

The literature reviewed highlights that implementing SSCM practices is complex given the number of players and stakeholder interests involved. SSCM requires a holistic view of the costs and benefits associated with specific environmental and social projects (Carter & Rodgers, 2008) across the supply chain. To achieve the high-level analysis, advanced analytics that utilize simulation and optimization models to understand the likely effects of every action are required ([31]. BDA, as demonstrated through the dimensions in Section 2.4, has technologies and analytics capabilities to collect data across the supply chain and perform analytics to develop insights that can achieve data-driven decision making regarding costs and benefits from a sustainability perspective. Thus, data analytics is required to understand the effect of each of the activities and classify those that support sustainability to be enhanced and the ones against sustainability to be eliminated or minimized. This finding is supported by [63] who emphasize the importance of predictive analytics in supply chain decision making regarding cost savings and competitiveness.

Research has shown that some social and environmental initiatives might harm the economic goal of the firm especially if there is no proper balance [44]. Proper analysis should be conducted to attain a balance among the environmental, social and economic goals [2]; subsequently, [31] highlights that BDA can offer a solution through predictive and descriptive analytics using simulation models, implying that BDA supports SSCM by evaluating the potential outcomes of the various social and environmental initiatives based on current data to shape the future with the aim of not allowing projects that can hurt a firm's bottom line and sustainability agenda. The analytics are likely to result in cost savings, reduced health and safety costs and shorter cycle times and enhanced reputation [2,63]. In addition, [9] argued that big data can improve the economic dimension of a firm through reduced operational costs, thus creating competitive advantages. Further, [25] also highlights that firms should build BDA capabilities which are known to support sustainable product development initiatives. Therefore, firms can implement specific sustainability initiatives from an informed perspective based on the insights obtained from big data analytics.

The application of integration and interoperable software for data sharing across the supply chain includes real-time analytics capabilities of data from internal and external sources (such as social media platforms which generate data at high velocity) to help draw insights based on mentions of firms from a particular supply chain. This helps to identify reputational damages early as well as detect disruptions and take mitigation measures to reduce the associated risks. In addition, data analytics on external data is also likely to help in risk management by identifying risks related to supply chain disruptions from protests, traffic, natural disasters, trade wars, political conflicts and even scarcity of raw materials as well as pandemics like COVID-19. As such, [9] supports that there is a relationship between big data analytics and sustainable supply chain management, whereby big data analytics tools promote the sustainability of manufacturing supply chains by transmitting

real-time actionable reports from the collected data. Thus, identifying supply chain risks earlier using insights from BDA and mitigating them promotes SSCM [2,63]. Figure 4 illustrates the likely results of the relationship from a BDA-powered sustainable supply chain management.

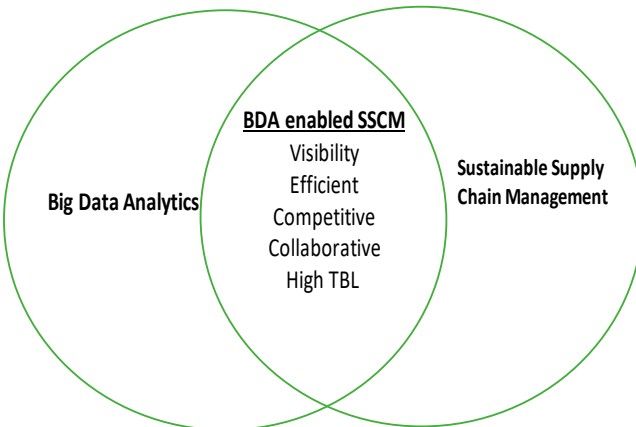

**Figure 4.** Conceptualized relationship between big data analytics and SSCM.

Based on the foregoing discussion, it is concluded that

*Big data analytics promotes sustainable supply chain management.*

The above claim is presented with certainty; however, arguments cannot be certain. As such, there is need to make them flexible by using qualifiers [59]. The qualifiers change the claim from being certain and rigid to flexible. Therefore, the stated claim becomes

**Claim:** *Big data analytics presumably promotes sustainable supply chain management in manufacturing supply chains.*

After stating the claim, it is important to present the evidence and facts (which are known as grounds in Toulmin's model) that support the claim [59]. The grounds can be statements of fact, statistics or expert opinions [59] that provide evidence and facts that big data analytics is positively associated with sustainable supply chain management. There is sufficient evidence that big data analytics can be applied successfully in supply chain management functions, such as sourcing [33], sourcing cost improvement and risk management [64], supplier selection [65], manufacturing and production planning [66], logistics planning and inventory management [67], demand management [54] and warehousing [53]. BDA has also been applied to supply chain management as a whole, as evidenced by [63], to achieve cost reductions. BDA can be applied to achieve data-driven SCM decision making, which also comprises smart manufacturing [13]. BDA offers better SCM integration capabilities, which can help overcome the challenges of traditional SCM problems, such as the bullwhip effect, supply demand mismatch, over production and ineffective response, by leveraging the insights from data analytics to minimize the supply disruption risks [9,13]. In addition, BDA can result in the improved economic dimension of SSCM through operational efficiency and strategic capability to create a competitive advantage [63]. BDA can be applied to improve business-to-business supply chain efficiency and risk management performance, which generally address the economic and environmental dimensions as well as the transparency facet of SSCM [2,62,68]. All the aforementioned studies have examined BDA and SCM for the purpose of cost reductions and achieving high customer responsiveness; this implies that BDA can also be applied in pursuance of the social and environmental goals of SSCM. A more specific study of BDA and SSCM was carried out by [9], who concluded that big data predictive analytics enhances integration and cooperation within the supply chain. Although [9] did not establish the specific relationship between BDA and SSCM, it can be gleaned that it is likely to enhance sustainability initiatives if well implemented. Therefore, the grounds are

**Grounds:** *BDA enhances supply chain management integration, which results in better risk management, transparency, sustainability organizational culture and integration of sustainability with corporate strategy. Thus, better economic, social and environmental performance is a result of BDA deployment.*

The envisaged positive relationship between BDA and SSCM is possible when firms within a supply chain collaborate and share relevant data with an objective of not only improving their economic well-being but also social and environmental well-being [13]. The level of integration of the various supply chain links is a key determinant to achieving SSCM through big data analytics; it is expected that firms should have an open relationship that allows transparency in data processing and analytics as well as in reporting supply chain activities [2,4]. Possessing BDA capability requires the right staff skills to effectively use BDA to attain the expected sustainability outcomes [25]. This implies that when BDA is applied diligently for decision making, it reshapes how supply chains collaborate, compete and implement sustainability [13]. Therefore,

**Warrant:** *Since BDA incorporates smart technologies in the form of IoT that detect, sense and continuously collect and analyse data across the supply chain for responsive and insight-based decision making, it is likely to increase transparency, improve risk management and encourage a sustainability culture with a supply chain. This results in reduced disruptions, emissions and unethical behaviour. The positive results of BDA deployment can be realized by examining data related to economic, social and environmental performance of firms, which will be sufficient indicator of SSCM performance.*

**Backing:** *the warrant presented above is supported because the dimensions of BDA, such as integration, analytics, data processing, security and the economic, promote SSCM, which results in benefits, such as end-to-end visibility, cost savings and supply chain competitiveness.*

The challenges to BDA implementation may be due to lack of understanding [31], inability to identify suitable data, low acceptance of BDA as way to improve sustainability, data security and privacy issues [69]. Lack of structural capabilities to handle and process big data can hinder effective application of BDA to support SSCM [13]. Effective BDA benefits can be obtained when all the supply chain partners have developed the required capabilities to handle and process big data for decision making—the capabilities include advanced analytic tools for capturing, storing and analysing data [17]. Lack of the stated capabilities might result in suboptimal utilization of BDA. Some of the factors that might invalidate the likely positive influence of BDA on SSCM include lack of top management support, limited financial strength and limited data security and privacy procedures. Weak system security of the BDA ecosystem might result in the unethical use of big data and uncertain return on investment, especially from a social and environmental perspective [17]. In addition, poor deployment of BDA may raise integrity issues about the data collected, thus dampening the likely insights based on the data [70]. Limited security can also be due to lack of sufficient technical infrastructural capabilities, which is also likely to result in suboptimal results from BDA [71]. Therefore, the rebuttal is

**Rebuttal:** *BDA is likely to promote SSCM in manufacturing supply chains unless the supply chain partners do not provide quality data or face some challenges in BDA implementation.*

## 5. Conclusions

This paper sought to identify the elements of BDA, propose the relationship between BDA and SSCM and determine the challenges of implementing BDA in manufacturing supply chains. An effective BDA system is identified as consisting of data processing capabilities, which incorporate the use of intelligent devices and software to collect heterogeneous data [34]. The data processing dimension should nurture transparency within the supply chain by promoting the accountability of all partners. This supports [37,43], who emphasize the essence of BDA promoting collaboration and cooperation so as to collect valuable data for decision making. BDA is also characterized as having high-level

reporting capabilities at near real time for timely decision making. The BDA reports should encourage engagement among the stakeholders through collaboration and cooperation, as also argued by [72]. BDA reports should include the social needs of the communities and stakeholders especially on emissions and capacity building. Another element of BDA was identified as security of the systems. Security includes data privacy, protection from internal and external cyberattacks and a legal framework to govern data privacy [45]. Effective security will be achieved when BDA has elaborate technical software technologies that support advanced features, such as fraud analytics for quick detection of unwanted activities [38,40,41]. BDA should perform advanced analytics, which includes descriptive, predictive and prescriptive analytics on both structured and unstructured data sets [48]. BDA should also allow for integration across the supply chain [33]. Integration allows for interoperability and flexibility, which results in seamless information sharing and quick response to demand networks as well as SSCM requirements [38]. Integration also ensures the compatibility of the various systems used across the supply chain so as to meet the objectives of BDA [41,51]. Finally, BDA should have an economic element, implying that it should be value generating to the business entities through cost reduction and improved customer service through the implementation of a sustainability culture.

Based on the reviewed literature, it can be gleaned that diligent implementation of BDA can promote SSCM through the facets identified by [2]. Therefore, the relationship is modelled as illustrated in Figure 5.

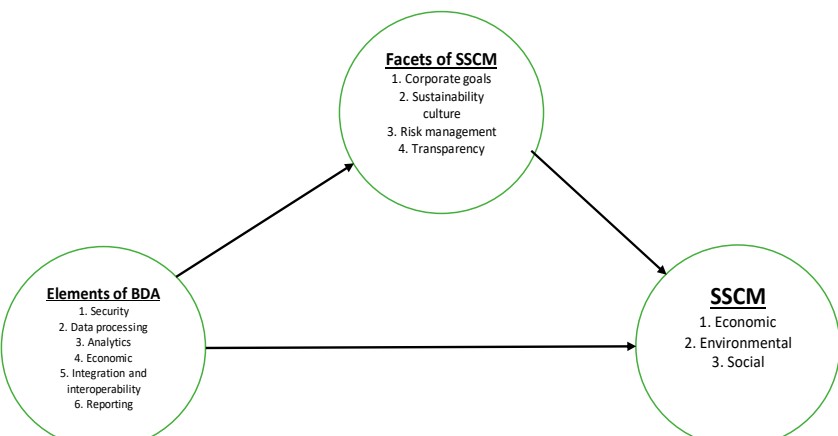

**Figure 5.** Proposed model of the relationship between BDA and SSCM.

As per Figure 5, the elements of BDA can be measured using the six dimensions, that is, data processing, data security, reporting, analytics, economic and integration and interoperability. The specific indicators of each of the dimensions can be adapted to the specific industry and context. The facets of SSCM are presented with the indicators that include corporate goals, sustainability culture and risk management. The indicators of sustainability are identified as environmental, social and economic, as presented by [7]. While Carter and Rodgers also provide some of the indicators for the SSCM dimension, the ones from [7] are preferred as they closely relate to the BDA elements.

The major challenges related to the implementation of BDA in manufacturing supply chains are identified as technological, human skills, top management and financial. The current study contributes to the field of sustainability by gleaning from literature the elements of BDA, establishing the relationship between BDA and SSCM, proposing a model that can be tested in future research and determining the challenges of implementing BDA from a manufacturing supply chain perspective.

This paper has its novelty in making an attempt to conceptualize and model the likely relationship between BDA and SSCM through facets such as corporate goals, sustainability culture, risk management and transparency. Prior studies have studied the two constructs separately. Further, the study has gleaned the dimensions or elements of the BDA concept

from the literature. This is likely to guide future research by providing metrics by which to measure BDA. The application of Toulmin's argumentation model in the literature discussion also contributes to the supply chain management literature and lays a foundation for future research.

Managers are informed that BDA has strong support across the academic and practitioner literature as a system as well as a concept that can help generate new insights based on big data collected across an enterprise or supply chain. BDA can help identify areas to improve in order to reduce operational costs, predict demand with accuracy, shape future demand with changing conditions and create supply chain resilience by minimizing disruptions and promoting all three dimensions of SSCM. Practitioners are informed that implementation of BDA is complex [5]: it requires top management support and a collaborative approach with other supply chain partners to promote SSCM goals. Managers are further provided with a likely checklist in terms of BDA dimensions to help ensure they have deployed an effective BDA system.

The paper is limited to the peer-reviewed studies that were consulted. A future research direction will be to test the proposed model empirically by collecting data from the manufacturing supply chains. It might be a study that collects data from different industries and countries and compares this data, given that BDA maturity would be at different levels especially between developed and developing countries. A study on the relationship between BDA and firm performance moderated by SSCM can add value to SCM literature.

**Funding:** This research received no external funding.

**Institutional Review Board Statement:** Not applicable.

**Conflicts of Interest:** The authors declare no conflict of interest.

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
