# Peer review of "Big Data Analytics in Sustainable Supply Chain Management: A Focus on Manufacturing Supply Chains"

_sustainability, doi:10.3390/su13137101_

Round 1

Reviewer 1 Report

The review report can be found in the attached file.

Author Response

Dear Editorial Team,

I am grateful for the reviewer 1 comments. They have helped improve the quality of the paper to a great extent. Please find the attached response to all the comments.

Thank you.

Author.

Reviewer 2 Report

Thank you for sending me for review the paper “Big data analytics in sustainable supply chain management: A focus on manufacturing supply chains”; Manuscript no. sustainability-1187745.

Presented methodology has a great potential in supply chain management and I am giving a support to the authors for investigation this topic. The strengths of this paper are: Relevant topic; Flow of the paper; and Explanation of the methods. However, the author(s) need to consider the following points as limitation or further scope for refining the paper:

- Introduction should be clearly stated research questions and targets first. Then answer several questions: Why is the topic important (or why do you study on it)? What are the research questions? What are your contributions?

- Citing of the references is not according to the Sustainability style.

- There are many typos in the text. For example, page 4, the end of the sentence “…stakeholders the sustainability initiatives they are engaged in for example pollution prevention activities so as to create awareness ().” What does mean “()”?

- Need to highlight the novelty of study in the introduction.

- Cleary define your motivations.

- All references cited in the text should be explained and discussed in a proper way. Try to compare these references and define the gap. Remove some old references published before 2017-2018. Some interesting references in the SCM field are missing. I suggest authors to read below interesting references: Giri, B. C., & Dey, S. (2020). Game theoretic models for a closed-loop supply chain with stochastic demand and backup supplier under dual channel recycling. Decision Making: Applications in Management and Engineering, 3(1), 108-125.

Biswas, S. (2020). Measuring performance of healthcare supply chains in India: A comparative analysis of multi-criteria decision making methods. Decision Making: Applications in Management and Engineering, 3(2), 162-189.

- Add flowchart of proposed methodology and follow that methodology in organization of your paper.

- Conclusion- Add more future scope. Do not use bullets or numerations in this section.

Author Response

Dear Editorial Team,

I am grateful for the reviewer 2 comments. They have helped improve the quality of the paper to a great extent. Please find the attached response to all the comments.

Thank you.

Author.

Round 2

Reviewer 1 Report

As the revised manuscript replied to the reviewer's previous comments. The reviewer recommends this paper to be accepted for publication.

Reviewer 2 Report

The authors have addressed the point of my concern. I am happy with their corrections. Hence, I would like to recommend this manuscript to be published.